# A Genome of Temperate *Enterococcus* Bacteriophage Placed in a Space of Pooled Viral Dark Matter Sequences

**DOI:** 10.3390/v15010216

**Published:** 2023-01-12

**Authors:** Ivan M. Pchelin, Pavel V. Tkachev, Daniil V. Azarov, Andrey N. Gorshkov, Daria O. Drachko, Vasily V. Zlatogursky, Alexander V. Dmitriev, Artemiy E. Goncharov

**Affiliations:** 1Scientific and Educational Center “Molecular Bases of Interaction of Microorganisms and Human” of the WCRC “Center for Personalized Medicine”, Institute of Experimental Medicine, 197022 Saint Petersburg, Russia; 2Smorodintsev Research Institute of Influenza, Ministry of Health of the Russian Federation, 197376 Saint Petersburg, Russia; 3Laboratory of Pathomorphology, Almazov National Research Centre, 197341 Saint Petersburg, Russia; 4Laboratory of Cellular and Molecular Protistology, Zoological Institute of the Russian Academy of Sciences, 199034 Saint Petersburg, Russia; 5Department of Invertebrate Zoology, Faculty of Biology, St. Petersburg State University, 199034 Saint Petersburg, Russia

**Keywords:** bacteriophage, *Enterococcus*, siphovirus, phylogeny, sequence space

## Abstract

In the human gut, temperate bacteriophages interact with bacteria through predation and horizontal gene transfer. Relying on taxonomic data, metagenomic studies have associated shifts in phage abundance with a number of human diseases. The temperate bacteriophage VEsP-1 with siphovirus morphology was isolated from a sample of river water using *Enterococcus faecalis* as a host. Starting from the whole genome sequence of VEsP-1, we retrieved related phage genomes in blastp searches of the tail protein and large terminase sequences, and blastn searches of the whole genome sequences, with matches compiled from several different databases, and visualized a part of viral dark matter sequence space. The genome network and phylogenomic analyses resulted in the proposal of a novel genus “Vespunovirus”, consisting of temperate, mainly metagenomic phages infecting *Enterococcus* spp.

## 1. Introduction

Temperate bacteriophages are estimated to constitute at least 20–50% of free phages in the human gut [1] or even outnumber the lytic ones [2]. While integrated in the bacterial chromosome, they offer their hosts the encoded adhesion factors, CRISPR arrays, toxins, and stress tolerance determinants, contributing therefore to bacterial adaptation [3]. Through packaging host DNA into virions and subsequent recombination within a new host, temperate phages facilitate horizontal gene transfer and drive bacterial evolution [4].

The composition of phage populations varies from person to person, but in adult individuals, it is generally stable at the time scale of days [5] and years [2]. The relative contribution of bacteriophages to the viral community of the gut is associated with inflammatory bowel disease, where the changes in the virome composition reflect alterations in the bacterial composition [6]. The bacteriophage abundance increases in individuals with Crohn’s disease [7]. Statistical analysis, including not only clinical or bacterial data, but also data on the fecal viral diversity, allows for a more accurate identification of patients with severe nonalcoholic fatty liver disease and fibrosis [8]. In alcoholic hepatitis, specific bacteriophage taxa were associated with an increased disease severity and 90-day mortality [9]. The virome signatures associated with colorectal cancer have a potential to be used for developing screening tools and predicting disease outcomes [10]. The mentioned studies relied on a phage classification at the levels of families [6,7], species [8,9], or species and genera [10] and illustrate why bacteriophage taxonomy is not only of a theoretical, but also of an applied significance.

In the studies on the human gut microbiome, the undescribed viruses may account for 75–99% of viral sequence reads [11]. The major part of this diversity is constituted by temperate bacteriophages. Due to their prevalence, a resistance to culturing techniques and sophisticated evolutionary patterns, these phages are often referred to as “the viral dark matter” [12]. The description and classification of bacteriophages lags behind the generation of metagenomic sequence data, hampering the interpretation of the experimental results [13,14].

Therefore, we aimed to describe a novel temperate Enterococcus phage VEsP-1. The taxonomic analysis of VEsP-1 and the related sequences from the GenBank substantiated a proposal for a new genus, “Vespunovirus”, represented mainly by metagenomic phage sequences. The genome network, calculated from sequences retrieved in nine sequential blast searches, allowed us to visualize the relationships between very dissimilar bacteriophages.

## 2. Materials and Methods

### 2.1. Isolation and Description of Enterococcus Phage VEsP-1

Bacteriophage isolation, host spectrum determination, transmission electron microscopy, and DNA sequencing were done with the use of protocols described in our previous publication [15]. Briefly, the bacteriophage VEsP-1 was isolated from a sample of river water enriched with the *E. faecalis* strain Serg. The same strain was used for bacteriophage propagation. The sample originated from Tô Lịch River (Hanoi, Vietnam). The host range was determined with the use of an in-house collection of 80 bacterial strains identified by MALDI-TOF mass spectrometry. The collection included 65 *Enterococcus* spp. strains and 15 isolates belonging to other bacterial genera, mainly *Staphylococcus* and *Streptococcus* (Appendix A). The lytic activity was determined on a bacterial lawn by a spot test with eight serial 10-fold dilutions of the phage stock preparation. The efficiency of plating values were calculated by dividing the phage titer determined on a bacterial lawn by a titer of phage on a lawn of *E. faecalis* Serg.

The size of viral particles (*n* = 8) was measured from transmission electron microscopy images. The probable natural range of the mean capsid diameter and tail length values was predicted by calculating the standard errors. DNA isolation was performed by the phenol/chloroform method with PEG 8000/NaCl precipitated bacteriophage preparation. The phage genome sequence was obtained with the use of the Illumina MiSeq platform (Illumina, San Diego, CA, USA). The library preparation was carried out with a Nextera XT DNA Library Preparation Kit (Illumina), resulting in 300 bp paired-end reads. Quality control was performed using FastQC 0.11.9 (https://www.bioinformatics.babraham.ac.uk/projects/fastqc/, accessed on 1 May 2020). A whole genome sequence was assembled de novo by SPAdes 3.13.0 [16] and deposited at the NCBI Nucleotide database with the accession number MZ333456. Coding sequences were predicted by the RAST genome annotation service (https://rast.nmpdr.org/rast.cgi, accessed on 7 December 2022) [17,18,19]. The tool was set to follow the RASTtk annotation scheme with frameshift correction. The possible spontaneous induction of VEsP-1 was excluded by searching the genome of the *E. faecalis* strain Serg (BioProject ID PRJNA884994) with the phage sequence.

### 2.2. Genome Network Analysis

To determine the position of VEsP-1 in the space of the related phage genomes, the VEsP-1 genome-centered sequence sample was obtained by performing three BLAST searches against the sequences of Caudoviricetes deposited in GenBank (NCBI:txid2731619). Two blastp searches were performed with the amino acid sequences of the terminase large subunit QYI86497.1 (TLS) and tail protein QYI86480.1 (TP) of VEsP-1. The retrieved accessions were used to query the NCBI Identical Protein Groups database and prepare the accession lists of the corresponding phage genome sequences. The obtained accession lists were pooled with the results of the blastn search with the VEsP-1 genome sequence MZ333456 (WGS) and were used to download sequence data from the NCBI Nucleotide database. To increase the sample size, the VEsP-1 genome-centered sample was pooled with sequences, obtained through similar searches with TLS, major capsid proteins and the WGS of Streptococcus phage P7954 (KY705280.1), and Brevibacillus phage Powder (KT151958.1). The searches were performed in July 2022. In all cases, every sequence producing a significant alignment was taken for a further analysis. The genomes shorter than 33,512 bp, 85% of the VEsP-1 sequence MZ333456.1, were considered to be incomplete and were thus rejected. For a taxonomic annotation, we downloaded the full NCBI Taxonomy entries associated with Nucleotide accessions in the XML file format with the use of Batch Entrez feature. To connect the sequence accessions to the taxonomic data, the associated taxonomy identifiers were retrieved using Entrez ESummary utility.

The structure of the sequence space in the general sample of the genomes was determined by a vContact2 0.11.3 analysis with the default parameters [20,21]. The calculations were carried out with the protein sequences predicted by prokka 1.14.6 [22]. The obtained pairwise distances were based on the probability of sharing a certain number of protein clusters between the genomes [20]. The resulting network file was visualized with the use of Cytoscape 3.9.1 [23] with edge weights interpreted as negative Log values. The group of VEsP-1-related viruses was predicted by vContact2 as a cluster of sequences with a statistically significant number of shared genes [20].

### 2.3. Phylogenomic Analysis

The phylogenomic analysis was performed with the VICTOR web service (https://victor.dsmz.de, accessed on 5 September 2022) [24]. The analysis included pairwise comparisons of the nucleotide sequences using the Genome-BLAST Distance Phylogeny (GBDP) method [25] optimized for prokaryotic viruses [24]. The resulting intergenomic distances were used to infer a balanced minimum evolution tree with branch support via FASTME, including SPR postprocessing [26] for each of the formulas D0, D4, and D6, respectively. Branch support was inferred from 100 pseudo-bootstrap replicates each. Taxon boundaries were estimated with the OPTSIL program [27], with the clustering thresholds described by Meier–Kolthoff and Göker [24] and the fraction of links required for a cluster fusion of 0.5 [28]. Intergenomic similarities were calculated using the VIRIDIC tool with the default parameters (http://rhea.icbm.uni-oldenburg.de/VIRIDIC/, accessed on 9 January 2023) [29]. Additional methods are described in the Appendix A [30,31,32].

### 2.4. Functional Analysis

The search for homologous protein-coding genes was performed with the CoreGenes 5.0 webserver (http://coregenes.ngrok.io/, accessed on 27 July 2022) using the Group Clustering: Bidirectional Best Hit Algorithm [33]. The prediction of capsid structure was done using the code described by Lee et al. on the basis of the sequence of the gene coding for a major capsid protein. The result was expressed as the T-number, which is a discrete index determining the possible capsid surfaces compatible with icosahedral symmetry and 1/60th of the estimated number of the MCP molecules needed to complete the head of phage particle [34]. The search for the known antibiotic resistance determinants and bacterial virulence factors was performed using the ABRicate 1.0.1 tool [35] and the databases ARG-ANNOT, CARD, MEGARes 2.00, NCBI AMRFinderPlus, Resfinder, and VFDB [36,37,38,39,40,41].

Bacteriophage protein immunogenicity prediction was done with the use of two web servers: VaxiJen 2.0 (http://www.ddg-pharmfac.net/vaxijen/scripts/VaxiJen_scripts/VaxiJen3.pl, accessed on 10 August 2022) [42,43] and VirVACPRED (https://virvacpred.herokuapp.com/, accessed on 10 August 2022) [44] with the thresholds recommended in the original publications. Both web servers are designed to perform alignment-free searches for protective immunogens with whole protein sequences as an input.

## 3. Results

### 3.1. Phenotype of Enterococcus Phage VEsP-1

The bacteriophage VEsP-1 was isolated from a sample of river water. It was able to produce plaques on four *Enterococcus faecalis* strains (11% of tested *E. faecalis* isolates). The results of a spot test assay with other bacterial isolates were negative (Appendix A). Transmission electron microscopy revealed viral particles with a siphovirus morphology, possessing a long flexible tail and an icosahedral head (Figure 1). The average head size was 50.3 ± 0.6 nm and the average tail length was 148.1 ± 11.6 nm. The presence of the gene coding for integrase implied the temperate lifestyle.

As a result of DNA sequencing, 1,399,584 paired-end reads were obtained. The viral genomic sequence was assembled de novo into a single contig with 60× coverage and was manually corrected. The length of the genome was 39,221 bp with the GC content at 35.1%. The genome contained 63 protein-coding genes and 1 tRNA gene.

### 3.2. Taxonomic Analysis

The closest match to the obtained whole genome sequence of VEsP-1 was the metagenomic sequence of bacteriophage cts681 (BK058556.1) with a 53% similarity to the VEsP-1 genome as determined by the multiplication of the percent identity by the percent coverage. The closest genome with known taxonomic position had a 14% similarity and belonged to the Enterococcus phage phiFL3B, a member of the genus *Phifelvirus* (GQ478087.1). Consequently, bacteriophage VEsP-1 could not be identified at the species or genus levels.

Aiming at elucidating the position of the VEsP-1 genome in the space of the related sequences, we performed a genome network analysis. The preliminary vContact2 analysis was done with the results of the BLAST searches with the VEsP-1 whole genome sequence, tail protein, and terminase large subunit sequences. The analysis placed the VEsP-1 genome in a cloud of the phage sequences. To increase the sample size, the BLAST searches were repeated with the sequences of the topologically close Streptococcus phage P7954 (*Brussowvirus*) and Brevibacillus phage Powder (*Jimmervirus*, not shown). The two genomes shared homologous sequences coding for a terminase small subunit and portal protein with VEsP-1. During the sequential BLAST searches, we obtained an increase in the sample size from 1214 to 6739 sequences (at a factor of 5.6) moving from VEsP-1 WGS to the general sample.

All nine sequences used in the BLAST searches to prepare the general sample contributed a noticeable number of unique matches (Figure 2).

The vContact2 network calculated from our general sample of genomes (*n* = 6739) contained 6711 nodes; the main interconnected part counted 6359 nodes or 94% genomes of the original sample, including representatives of 112 bacteriophage genera. The only revealed group of genomes with ICTV-accepted family names belonged to *Herelleviridae* (Figure 3). The occasional sequences with family annotations in the main part of the network belonged to the genera *Eclunavirus*, *Webervirus* (*Drexlerviridae*, one sequence for each genus), *Cellubavirus* (*Assiduviridae*, one sequence), and *Baltivirus* (*Pachyviridae*, two sequences). In the smaller fragmented parts of the network, there were two *Pleakleyvirus* genomes (*Zierdtviridae*) and one *Nohivirus* genome (*Autographiviridae*). In the results of the vContact2 cluster analysis, the VEsP-1-containing group comprised nine members (Appendix A). The genus confidence score for the respective subcluster was calculated at 0.8889 (Appendix A). To find whether it is possible to interpret this group from a taxonomic point of view, we performed a phylogenomic analysis with VICTOR.

To prepare the VICTOR dataset, we selected the genomes (1) belonging to bacteriophages from the VEsP-1 cluster obtained during vContact2 analysis; (2) coding for the related tail protein sequences, forming a branch with a VEsP-1 cluster on a preliminary TP tree, together with the closest outgroup (Appendix A); and (3) belonging to genera, topologically close to VEsP-1 on the vContact2 phylogenetic network and at the same time containing at least one homologous sequence with VEsP-1 cluster members. The following bacteriophages were initially selected as topologically close but did not share homologous genes with VEsP-1 cluster members: *Colneyvirus*, *Fernvirus*, *Lacnuvirus*, *Lilyvirus*, *Lubbockvirus*, *Moineauvirus*, *Phifelvirus*, *Spizizenvirus*, and *Sukhumvitvirus*.

The resulting dataset for VICTOR included phage sequences sharing at least one homologous sequence with the VEsP-1 cluster. The hosts of the analyzed bacteriophages belonged to 11 Firmicutes genera, including two genera from the order Eubacteriales (*Clostridioides* and *Clostridium*), four genera from the order Bacillales (*Bacillus*, *Brevibacillus*, *Listeria*, and *Paenibacillus*), and five genera from the order Lactobacillales (*Enterococcus*, *Lactobacillus*, *Leuconostoc*, *Oenococcus,* and *Streptococcus*).

The genome-BLAST distance phylogenomic trees inferred using the formulas D0, D4, and D6 had an average support of 65%, 8%, and 70%, respectively. At the genus level, the OPTSIL clustering yielded eighteen (D0), fourteen (D4), and eighteen (D6) taxonomic hypotheses. Since the D4 tree had a low support and the supposed division of the dataset into genera with D0 and D6 formulas was identical, we considered only the D0 results (Figure 4). Ten of the 15 genus hypotheses matched the current ICTV-accepted genera. *Anamdongvirus*, *Jarrellvirus*, *Pleeduovirus,* and *Seongbukvirus* were represented by one sequence each. For the remaining genera reproduced in the OPTSIL analysis, we calculated intergenomic similarity values: *Harrisonvirus*, 50–98%; *Leicestervirus*, 45–100%; *Yongloolinvirus*, 49–87%; *Sozzivirus*, 79–82%; *Cequinquevirus*, 78–88%; and *Brussowvirus*, 54–88%. Here, four ICTV-accepted genera united viral genomes with a sequence similarity below the canonical 70% threshold. The Lactococcus phage BK5-T (*Sandinevirus)* and Lactococcus phage bIL285 (not classified) belonged to the same hypothetical genus. The remaining four OPTSIL genus hypotheses united 2–4 accepted genera. In all cases, the genera were monophyletic and in every accepted genus, the members belonged to one individual genus hypothesis. The VEsP-1 cluster genomes formed a highly supported monophyletic clade and belonged to an independent genus hypothesis, providing further evidence for a description of a taxonomic group (Figure 4). The nucleotide sequence similarity between the nine representative genomes was 17–77%.

### 3.3. The Proposed Genus “Vespunovirus”

The nine “Vespunovirus” genomes belonged mainly to uncharacterized bacteriophages found in metagenomic studies [47]. Since all VEsP-1-related viruses originated from fecal samples, VEsP-1 itself probably has a fecal origin. The genomes ranged in size from 35,370 to 42,320 bp and coded for 51–66 proteins (Table 1). The capsid structure was uniform, with T-numbers predicted at 7.0. The GC content varied from 33.5 to 37.9%. The group was monophyletic on a phylogenetic tree calculated from major capsid protein sequences (Appendix A). The CoreGenes analysis revealed 16 conservative coding sequences, including the portal protein gene in the DNA packaging module, the gene coding for integrase, and 14 genes in the structural module (Figure 5).

In “Vespunovirus”, several different terminase families were represented (Table 1). Six representatives possessed the XtmB super family terminase, sharing a 74–82% similarity with the sequence YP_004301277.1 of the Brochothrix phage NF5, suggestive of the P22-like headful packaging mechanism [48]. The metagenomic phages ct40X5 and ctCNj1 had COG5362 super family terminases homologous to the sequence NP_112694.1 of the Lactococcus phage TP901-1. In these two cases, a pac site-dependent packaging mechanism may be expected [49]. The TLS of bacteriophage ctyrd11 was 53% similar to the entry YP_009195762.1 from the genome of the Clostridium phage phiCD505 and belonged to the DEAD-like helicases superfamily. For the Clostridium phage phiCD505, a T4-like packaging mechanism was predicted [50].

The “Vespunovirus” genomes lacked RNA polymerase. There was some degree of variability in the genome structures. For example, the ctCNj1 genome coded for one terminase subunit unlike other “Vespunovirus” genomes with two TER subunits. In the ctCNj1 genome, the putative DNA replication and recombination module genes alternated with lysogeny control genes. In the cthD61 genome, the DNA replication and recombination module was not interrupted by HNHc endonuclease, unlike the closely related SEsuP-1 and ct40X5 genomes (Figure 5).

Analyzing the relationships of what will later become closely related but separate genera *Slepowronvirus* and *Psavirus*, Denes et al. compared the lengths of tail tape measure proteins [52]. In our cluster of VEsP-1-related viruses, the genome of the bacteriophage ctCNj1 encoded the tape measure protein of 849 amino acids. In the metagenomic phage ctyrd11, the protein consisted of 938 residues and in other “Vespunovirus” members, the length of the protein was 965–974 amino acids. The shortest tape measure protein counted 87% amino acids of the longest one. In *Slepowronvirus*, we found the same ratio of 0.87 between the lengths of the tail tape measure protein of Listeria phage LP-HM00113468 (1420 amino acids) and the protein of the Listeria phage B025 (1640 amino acids). In *Psavirus*, the Listeria phage LP-030-2 and Listeria phage PSA had tail tape measure proteins of the same length of 1026 amino acids.

Using the ABRicate tool, we did not find antibiotic resistance determinants and bacterial virulence factors in the “Vespunovirus” genomes. However, they contained a number of genes potentially contributing to a bacterial adaptation. The most common genes of this kind coded for the ImmA/IrrE family metallo-endopeptidases and putative HNHc nucleases, each found in six genomes. In *Bacillus*, ImmA-dependent proteolysis of ImmR repressors is considered to be a conserved mechanism for regulating a horizontal gene transfer [53]. In *Geobacillus*, the ImmA/IrrE family protein is a part of the toxin-antitoxin ToxN/AbiQ system [54]. HNHc nuclease domain-containing proteins may contribute to a bacterial virulence by diminishing the reactive oxygen species, and also by participating in the DNA repair [55]. Each other potential bacterial adaptation factor occurred in one or two genomes. The XhlA protein is a bacterial cell surface-associated hemolysin. It is capable of lysing rabbit and horse erythrocytes [56]. The Gp157 family proteins may contribute to a bacterial resistance to the bacteriophages [57]. In *Escherichia coli*, the cold shock proteins of the CspA family are involved in a cold-shock acclimation and are probably important for growth under optimal conditions [58]. Ferrochelatase catalyzes the insertion of the ferrous form of iron into protoporphyrin IX in the heme synthesis pathway. The LysM domain containing the proteins of *Enterococcus faecium* have unknown cellular functions. However, a transcriptomic analysis of *E. faecium* in an animal infection model revealed a significant increase in the number of corresponding transcripts, suggesting a role in the infection process [59]. Glucose-6-phosphate dehydrogenase is a part of the SoxR oxidative stress regulon of *E. coli* [60]. The LemA protein family was characterized in the genus *Pseudomonas*, where its members were identified as transmembrane histidine protein kinase sensor-regulators, involved in the formation of lesions in the host [61].

The revealed conservation in the structural module of the “Vespunovirus” genomes led to a question about the immunological properties of its proteins. To answer this question and provide a basis for a discussion on the significance of bacteriophage taxonomy at the genus level, we performed a computational prediction of immunogenicity in three related genera of bacteriophages sharing a conserved integrase sequence (Figure 6). In all three genera, there were confidently predicted immunogenic structural proteins. There was no clear difference in the overall distribution of the predicted immunogens according to the genus. However, unlike the major capsid proteins of *Sozzivirus* and *Psavirus* isolates, the MCPs of the “Vespunovirus” members apparently were not immunogenic.

## 4. Discussion

Until recently, the demarcation of species and genus-level groups in bacteriophage taxonomy relied on nucleotide sequence similarity thresholds and molecular phylogenetic analysis. The bacteriophages sharing at least a 95% identity between the whole genome sequences were classified as the same species. The threshold for the genera was set at 70% [63]. Making the use of strict thresholds problematic, the genomes of most temperate bacteriophages have regions of a high similarity within the pairs of otherwise very dissimilar genomes, resulting in reticulate patterns of evolution [64,65,66]. Consequently, the current ICTV guidelines suggest the use of group-specific demarcation criteria rather than rigid similarity thresholds (https://ictv.global/taxonomy/about, accessed on 13 September 2022) [67].

Since the applicability of species rank to mosaic bacteriophages has been put under question [20,68], we refrain from proposing a species for the described bacteriophage VEsP-1. The revealed group of VEsP-1-related viruses shows a remarkable mixture of traits usually seen at different levels of phage taxonomy. On the one hand, clustering in vContact2 and VICTOR analyses is definitely evidence of a genus group. The nine viruses are highly likely to share hosts belonging to the same genus *Enterococcus*. The degree of differences in the lengths of the tape measure proteins support the genus status of the cluster. Additionally, the group was monophyletic in a signature gene phylogenetic analysis. On the other hand, three different packaging mechanisms as well as a slightly varying genomic organization favor a description of a family-level taxon. The number of 16 shared genes may also be interpreted as a family-level trait when compared to 31 shared genes in *Wbetavirus* [69]. Due to the limited number of representatives and the apparently minor significance of the proposed taxon, we suggest a grouping of the genus level. Though the intergenomic similarities between “Vespunovirus” members are low, there are at least four related ICTV-accepted genera with minimum nucleotide sequence similarity values of the genomes below the 70% threshold. Nearly all conserved in the “Vespunovirus” genes were structural, as it was earlier found in *Brussowvirus* and *Moineauvirus* members [70,71]. Therefore, we speculate the mentioned viral genera may have a significance for immunological studies.

Our two-step pooling approach to preparing a nucleotide sequence sample allowed us to increase the number of analyzed sequences by a factor of 5.6. The vContact2 analysis presented here predicted statistically significant similarities with at least one another genome in 94% of the genomes of our general sample. Therefore, with the mosaic viruses in question, the pooling results of several BLAST searches may seem reasonable.

Through the sequential steps of a significant similarity between the neighboring genomes, our vContact2 network connected very dissimilar phages with the hosts from at least three different orders of Firmicutes. The obtained network depicts otherwise elusive relationships between bacteriophages sharing no homologous genes and possessing different genome organizations and can be thought of as a kind of substitute to higher taxonomic ranks in the studied part of viral dark matter. Probably, the meaning behind this would be in organizing the inventory of existing taxonomic names. It remains to be seen whether functional traits can be ascribed to phage groups of this magnitude.

## Figures and Tables

**Figure 1 viruses-15-00216-f001:**
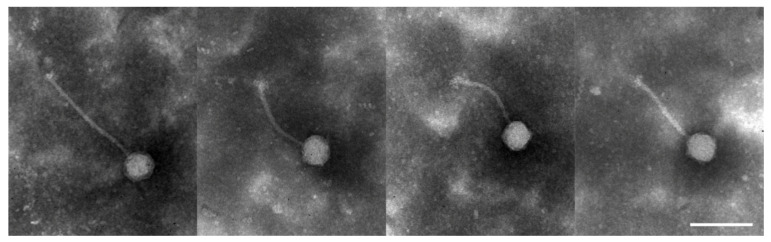
Transmission electron microscopy of Enterococcus phage VEsP-1. Scale bar, 100 nm.

**Figure 2 viruses-15-00216-f002:**
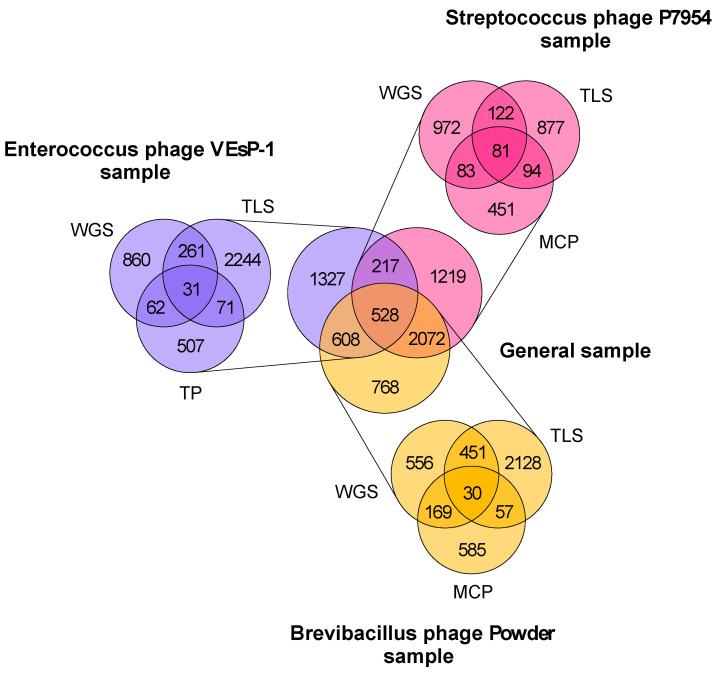
Sequence sample preparation. The numbers of whole genome sequences retrieved in nine BLAST searches with bacteriophage whole genome sequences (WGS), amino acid sequences of major capsid protein (MCP), terminase large subunit (TLS), and tail protein (TP) are shown. The general sample was obtained by pooling the results of the searches. The diagrams were visualized in VennDiagram 1.7.3 package [45] for R 4.1.2 [46].

**Figure 3 viruses-15-00216-f003:**
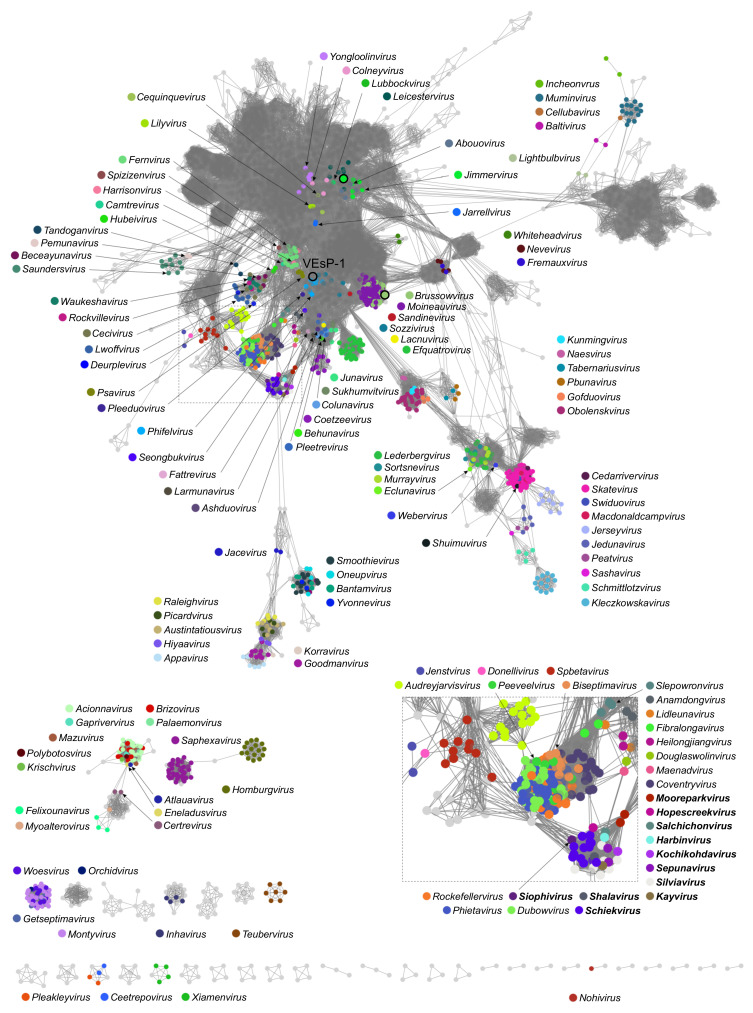
Network analysis of bacteriophage genomes. The inset shows a part of the network containing *Herelleviridae* members (given in boldface). The circles with outer boundaries show starting points for BLAST searches: Enterococcus phage VEsP-1, Streptococcus phage P7954 (*Brussowvirus*), and Brevibacillus phage Powder (*Jimmervirus*). The network was calculated in vContact2 0.11.3 and visualized with the use of Cytoscape 3.9.1 program.

**Figure 4 viruses-15-00216-f004:**
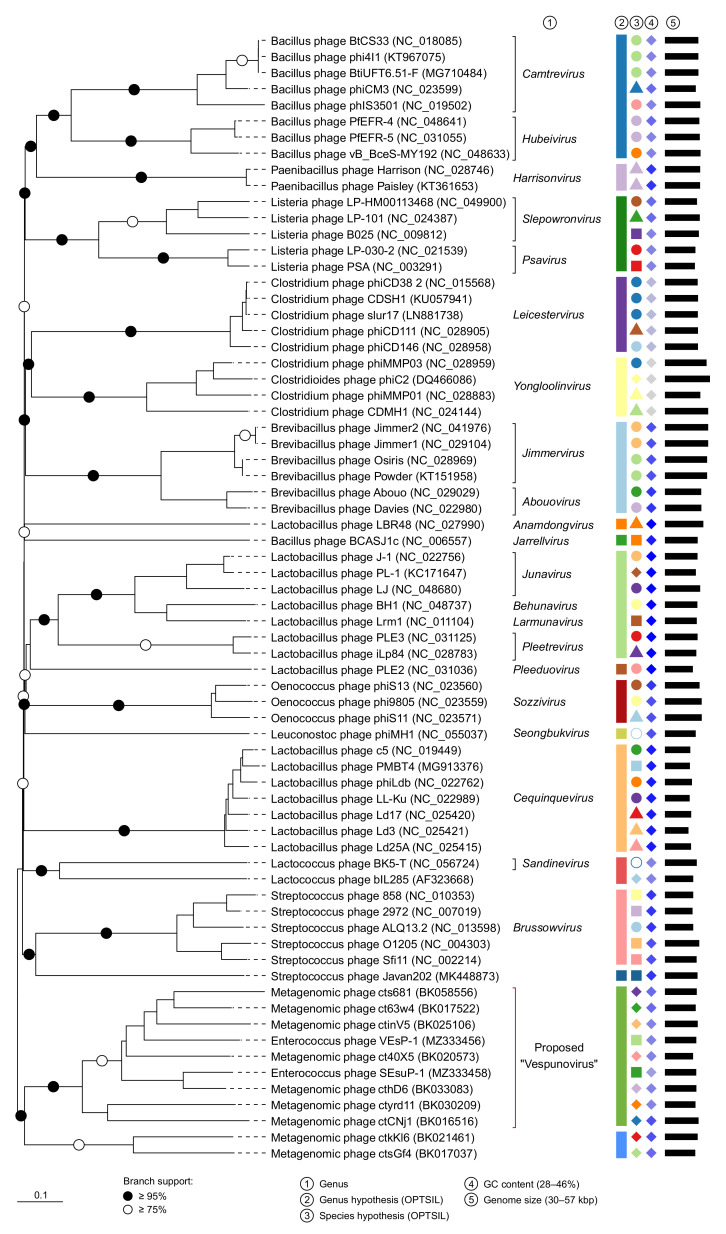
VICTOR phylogenomic tree. The group of VEsP-1-related bacteriophages predicted in vContact2 analysis belong to the same “Vespunovirus” genus hypothesis.

**Figure 5 viruses-15-00216-f005:**
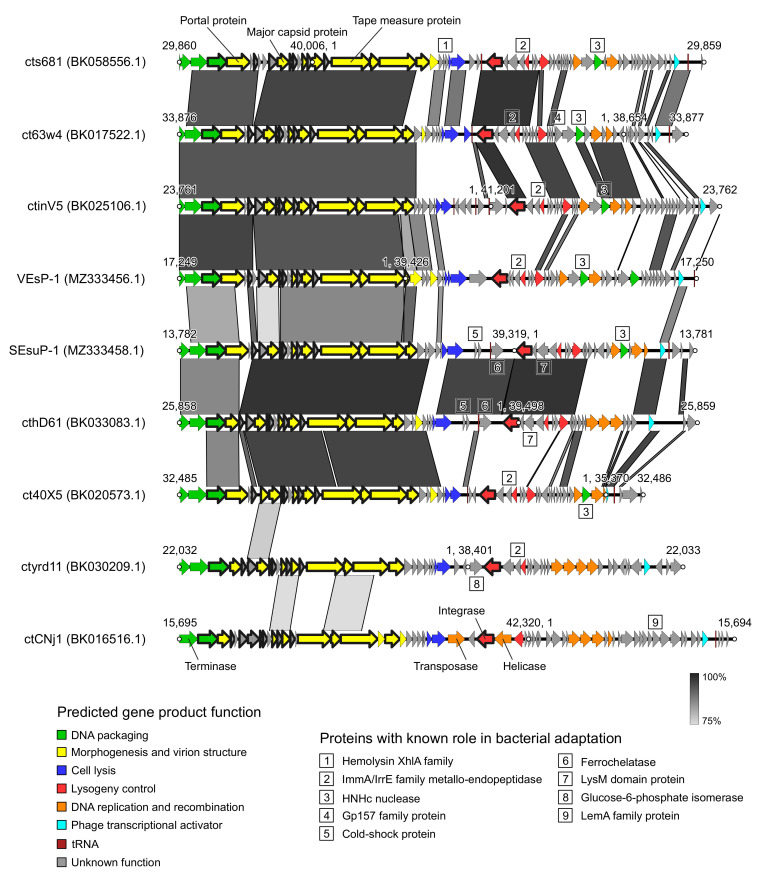
Genome structures in the proposed “Vespunovirus” genus. The conserved protein coding sequences are marked with a thick outline. The shades of grey show nucleotide sequence identity. The image was prepared using Easyfig 2.2.5 [51] on the basis of Prokka 1.14.6 annotations.

**Figure 6 viruses-15-00216-f006:**
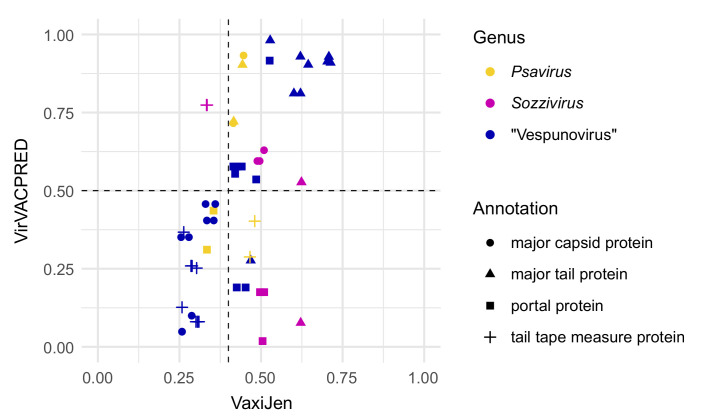
Probability of immunogenic potential of bacteriophage structural proteins in “Vespunovirus”, *Psavirus,* and *Sozzivirus* members. The thresholds for probability values are shown as dashed lines. The proteins were homologous within the genera, but not between the genera. The plot was visualized in ggplot2 3.3.5 package [62] for R 4.1.2 [46].

**Table 1 viruses-15-00216-t001:** The genomes belonging to the proposed genus “Vespunovirus”.

Phage Name	Sample Location	Sample Type	GenBank Accession	Length, bp	ORFs	tRNAs	GC Content, %	Terminase Large Subunit	Top Bacterial BLAST Hit; Query Cover/Identity, %
cts681	ND *	Human feces	BK058556	40,006	63	2	35.4	XtmB	*Enterococcus faecalis*; 71/99
ct63w4	Estonia	— “ —	BK017522	38,654	59	2	35.4	XtmB	*E. faecalis*; 72/96
ctinV5	Denmark	— “ —	BK025106	41,201	66	3	35.0	XtmB	*E. faecalis*; 69/95
VEsP-1	Vietnam	River water	MZ333456	39,221	63	1	35.1	XtmB	*E. faecalis*; 48/95
SEsuP-1	Russia	Human feces	MZ333458	39,319	51	2	33.5	XtmB	*E. faecalis*; 96/100
cthD61	USA	— “ —	BK033083	39,498	54	1	34.1	XtmB	*E. faecalis*; 99/100
ct40X5	Russia	— “ —	BK020573	35,370	52	2	33.9	COG5362	*E. faecalis*; 100/100
ctyrd11	Ethiopia	— “ —	BK030209	38,401	52	0	33.8	DEXDc	*E. hirae*; 83/98
ctCNj1	USA	— “ —	BK016516	42,320	66	1	37.9	COG5362 **	*E. avium*; 23/100

* ND, no data; ** in the genome, there is one terminase-coding gene.

## Data Availability

The whole genome sequence of the Enterococcus phage VEsP-1 was deposited in the NCBI Nucleotide database under the GenBank/ENA/DDBJ accession number MZ333456.

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
