# Peer review of "A Genome of Temperate Enterococcus Bacteriophage Placed in a Space of Pooled Viral Dark Matter Sequences"

_viruses, 2023, doi:10.3390/v15010216_

Round 1
Reviewer 1 Report (New Reviewer)
The manuscript by Pchelin et al describes isolation of the temperate Enterococcus phage VEsP-1 isolated from river water which demonstrated lytic ability on a relatively small subset (~10%) of E. faecalis strains tested. VEsP-1 was shown to have siphovirus morphology and a ~39 kb genome. The authors then performed a series of bioinformatic harvesting work to identify and retrieve phages similar to VEsP-1 through BLAST searching against the whole genome nucleotide sequence, tail protein, large terminase subunit and then used similar conditions on two related phages Streptococcus phage P7954 and Brevibacillus phage Powder to identify a total of 6739 phage genomes for subsequent network analyses. After determining a group of 9 phage members (most sourced from metagenomic studies) the authors performed VICTOR whole genome phylogeny analysis to further analyse these phages. The authors then group these phages into a proposed genus, Vespunovirus, and discuss various genome features. The manuscript is well written, the figures are clear, however I have concerns about the authors determination that these phages belong to a single genus and subsequent discussion that group-specific demarcation criteria are currently recommended rather than nucleotide similarity thresholds (95/70%).
It is my understanding that the 95% species and 70% genus nucleotide similarity thresholds for phage classification still remain the standard in ICTV classification - please see a 2021 publication from members of the ICTV bacterial viruses subcommittee in Viruses, https://doi.org/10.3390/v13030506. My interpretation of that publication is that 70% nucleotide similarity remains the gold standard cut-off for genera except for “edge-cases” - presumably when intra-genus phages are >70% similarity with some (but not all) phages within the proposed genera and can be admitted to said genus with additional evidence (i.e. conservation of signature genes etc.). The concept of edge cases likely exists because the ICTV has recently moved away from the idea of type species (see another 2021 paper https://doi.org/10.1007/s00705-021-05156-1) and prefer genera to represent a closely related group of phages (i.e. all members are pairwise within ~70% similarity of each other) rather than what's used elsewhere (i.e. bacteria) where a candidate needs to be within 95/70% of the type strain/species to be included in the respective species/genus.
I believe in this case the nucleotide similarity between the 9 proposed members of Vespunovirus drops too far below the currently accepted thresholds (<20% pairwise similarity for some members) to be considered a singular genus. I would suggest the authors revisit the more current literature indicated above and make a judgement from there. The authors could alternatively simply refer to these 9 phages as the Vespunovirus group (rather than genus) to avoid the specifics associated with taxonomic classification if they wish. I think the authors should also perform pairwise nucleotide similarity between the phages of the proposed Vespunovirus genus (i.e. VIRIDIC) and I would also like to see information added about the sub-VC assignment of the 9 phages by vContact2, where the sub-VC’s would hint at suggested genera grouping of the 9 phages (which in my experience yields a similar but slightly different result to the 95/70% nucleotide similarity groupings). OPTSIL genus predictions by VICTOR in my experience are not congruous with current classification standards. This is very evident in Figure 4 where OPTSIL on various occasions groups multiple ICTV-classified genera into a single predicted genus (i.e. Junavirus, Behunavirus, Larmunivirus and Pleetrevirus phages are predicted as a single genus by OPTSIL).
Please briefly write about the sequencing and assembly of the VEsP-1 genome in the Results section, probably most appropriately just prior to section 3.2. As it currently stands, you have not mentioned the size of the VEsP-1 genome (39,221 bp) and thus the author needs search for the phage on Genbank to ascertain this. Also in regards to the Genbank accession, I would recommend referring to it simply as MZ333456 (instead of MZ333456.1), as the prior notation immediately links to most recent version of this accession, which happens to be MZ333456.2. This suggestion would apply to usage on lines 87 and 102 in the submitted manuscript.
The VEsP-1 phage genome has been gene-called very well and the annotations are also of generally high quality. I would just note that some of the virion structural genes can be further annotated with the use of something like Virfam which additionally identifies gp 11, 12, 13 and 14 as an adapter protein, head closure protein, neck protein, and tail completion protein, respectively (instead of hypothetical protein, ribosomal protein L23, hypothetical protein and hypothetical protein, respectively, in the current annotations).
Author Response
R3: I have concerns about the authors determination that these phages belong to a single genus and subsequent discussion that group-specific demarcation criteria are currently recommended rather than nucleotide similarity thresholds (95/70%).
It is my understanding that the 95% species and 70% genus nucleotide similarity thresholds for phage classification still remain the standard in ICTV classification - please see a 2021 publication from members of the ICTV bacterial viruses subcommittee in Viruses, https://doi.org/10.3390/v13030506. My interpretation of that publication is that 70% nucleotide similarity remains the gold standard cut-off for genera except for “edge-cases” - presumably when intra-genus phages are >70% similarity with some (but not all) phages within the proposed genera and can be admitted to said genus with additional evidence (i.e. conservation of signature genes etc.). The concept of edge cases likely exists because the ICTV has recently moved away from the idea of type species (see another 2021 paper https://doi.org/10.1007/s00705-021-05156-1) and prefer genera to represent a closely related group of phages (i.e. all members are pairwise within ~70% similarity of each other) rather than what's used elsewhere (i.e. bacteria) where a candidate needs to be within 95/70% of the type strain/species to be included in the respective species/genus. I believe in this case the nucleotide similarity between the 9 proposed members of Vespunovirus drops too far below the currently accepted thresholds (<20% pairwise similarity for some members) to be considered a singular genus. I would suggest the authors revisit the more current literature indicated above and make a judgement from there. The authors could alternatively simply refer to these 9 phages as the Vespunovirus group (rather than genus) to avoid the specifics associated with taxonomic classification if they wish.
A: Before formulating the final interpretations of the results we noticed this discrepancy between the ICTV website and the publication by Turner et al. (2021). We honestly tried to follow up-to-date guidelines in an unbiased manner and decided to prefer the information from the website. This is because scientific communication through the website seemed to us to be more flexible and more authoritative. Please also see examples of ICTV accepted genera with minimum sequence similarity below canonical 70% threshold: Harrisonvirus, 50%; Leicestervirus, 45%; Yongloolinvirus, 49% and Brussowvirus, 54%. Finally, we believe the main idea of supraspecific taxa is in organizing the knowledge about multiple representatives.
R3: I think the authors should also perform pairwise nucleotide similarity between the phages of the proposed Vespunovirus genus (i.e. VIRIDIC)
A: The VIRIDIC calculations were performed, described and discussed in the MS.
R3: I would also like to see information added about the sub-VC assignment of the 9 phages by vContact2, where the sub-VC’s would hint at suggested genera grouping of the 9 phages (which in my experience yields a similar but slightly different result to the 95/70% nucleotide similarity groupings).
A: We attached the files with the results of vContact2 analysis to the manuscript as Supplementary Data 1 and 2. Genus Confidence Score for the “Vespunovirus” subcluster VC_302_0 containing nine members was calculated at 0.8889 (Data S2). This information was introduced to the paragraph below Figure 2.
R3: OPTSIL genus predictions by VICTOR in my experience are not congruous with current classification standards. This is very evident in Figure 4 where OPTSIL on various occasions groups multiple ICTV-classified genera into a single predicted genus (i.e. Junavirus, Behunavirus, Larmunivirus and Pleetrevirus phages are predicted as a single genus by OPTSIL).
A: As far as we know, VICTOR software is still among the recommended taxonomical tools. We performed three searches with Google Scholar, using the following queries: "OPTSIL" "genus" "limitation"; "OPTSIL" "genus" "usability"; "OPTSIL" "genus" "accuracy". In either case, we did not manage to find sources with criticism regarding the performance of the algorithm.
R3: Please briefly write about the sequencing and assembly of the VEsP-1 genome in the Results section, probably most appropriately just prior to section 3.2. As it currently stands, you have not mentioned the size of the VEsP-1 genome (39,221 bp) and thus the author needs search for the phage on Genbank to ascertain this.
A: In the Section 3.1, the sequencing and assembly of the VEsP-1 genome were described, as requested. The size of the genome was mentioned.
R3: Also in regards to the Genbank accession, I would recommend referring to it simply as MZ333456 (instead of MZ333456.1), as the prior notation immediately links to most recent version of this accession, which happens to be MZ333456.2. This suggestion would apply to usage on lines 87 and 102 in the submitted manuscript.
A: Corrected according to the suggestion.
R3: The VEsP-1 phage genome has been gene-called very well and the annotations are also of generally high quality. I would just note that some of the virion structural genes can be further annotated with the use of something like Virfam which additionally identifies gp 11, 12, 13 and 14 as an adapter protein, head closure protein, neck protein, and tail completion protein, respectively (instead of hypothetical protein, ribosomal protein L23, hypothetical protein and hypothetical protein, respectively, in the current annotations).
A: The entries QYI86488.1 adaptor protein, QYI86487.1 head closure protein, QYI86486.1 neck protein and QYI86485.1 tail completion protein were updated, as recommended.
A: We also checked the manuscript once again and fixed a number of minor errors. The list of amendments included text on Figure 2.
Reviewer 2 Report (Previous Reviewer 2)
The authors have revised the questions raised by the reviewer and it can be accepted.
Author Response
Thank you.
Reviewer 3 Report (Previous Reviewer 1)
I have no further criticisms.
Author Response
Thank you.
Round 2
Reviewer 1 Report (New Reviewer)
Pchelin et al. have included the additional analyses recommended, however may I ask that the results of the intergenomic similarity be included in the manuscript or in the supplementary file, as it currently stands the authors simply state in the discussion "Though intergenomic similarities between “Vespunovirus” members are low, there are at least four related ICTV accepted genera with minimum nucleotide sequence similarity values of the genomes below the 70% threshold". Apologies if its already there and I've somehow missed it.
Regarding the OPTSIL predictions in the VICTOR analysis, I agree with the authors there have not neccssarily been any published reports indicating disagreement with its predictions. As evidence, I simply reiterate that the results within the authors own VICTOR analysis indicates the OPTSIL predictions do not always correlate with current ICTV classifications as it tends to group phages from multiple ICTV genera (i.e. Junavirus, Behunavirus, Pleetrevirus) into a single predicted genus. That being said, the authors believe they have solid grounds for inclusion of the phages into the singular genus of Vespunovirus. I encourage the authors to submit a proposal to the ICTV.
Congratulations.
Author Response
Dear Reviewer,
Many thanks for your valuable suggestions.
In the MS, please find the following description of nucleotide sequence similarity analysis: "Intergenomic similarities were calculated using VIRIDIC tool with default parameters (http://rhea.icbm.uni-oldenburg.de/VIRIDIC/, accessed on 09 January 2023)[29]." (Section 2.3); "For the remaining genera reproduced in OPTSIL analysis we calculated intergenomic similarity values: Harrisonvirus, 50-98%; Leicestervirus, 45-100%; Yongloolinvirus, 49-87%; Sozzivirus, 79-82%; Cequinquevirus, 78-88% and Brussowvirus, 54-88%. Here, four ICTV accepted genera united viral genomes with sequence similarity below canonical 70% threshold.", "The nucleotide sequence similarity between the nine representative genomes was 17-77%." (Section 3.2).
We agree that the results of the study are somewhat confusing, given that "The revealed group of VEsP-1-related viruses shows a remarkable mixture of traits usually seen at different levels of phage taxonomy." We painted the picture, but of course, the true decision about the taxonomic status of the described assemblage will be made by ICTV members.
Best wishes,
the Authors
This manuscript is a resubmission of an earlier submission. The following is a list of the peer review reports and author responses from that submission.
Round 1
Reviewer 1 Report
The manuscript reports the sequence of Enterococcus phage VEsP1, and provides a variety of characterizations indicating that it occupies an uncharacterized section of tree space and can be clustered into a Genus-level taxon with some metagenomic sequences.
The manuscript is correct in its major assertions. You can also see just from blast searches of bacterial genomes that this is a major temperate phage family in Enterobacteria with more distant homologs in Bacillus, Clostridium and some other Gram positive taxons. And it is extremely distant from any other characterized phage. This is certainly a worthwhile manuscript.
My major criticism is that the underlying GenBank file is full of problems, which causes confusion in the manuscript.
Problem 1 with the GenBank file: It is misassembled. There is a repeat of 129..333 with the end (39222..39426). That never happens in phages. This is the sort of thing that happens with either a recombinant amplificate that generates false paired end reads, or there was an insertion or deletion in the phage genome as it was growing up leading to a heterogeneous sequence. In either case, reads are produced that the assembler can't correctly assemble, and a human has to actually look at it. In this case the sequence from 129-333 should be deleted, and the right end should be contiguous with coordinate 334 in the circular form of the sequence. One would then typically circularly permute the sequence to put the end in a convenient place for reporting. You can confirm this arrangement because the gene annotated through this garbled region, QYI86480, if retranslated after the fix I indicate above it will magically recover its missing C-terminal segment in the blast searches. If you look at the sequence from 1..129 more carefully, you may be able to make the case that it is an insertion sequence of some kind making the phage heterogeneous. The version I point out above matches the rest of your family. So most authors would report only that, or at most make a minor note that there was a variant sequence in the mix with an insertion in it, assuming they could reconstruct exactly what the variant was. This is relevant to the manuscript because this gene was used in the classification analysis. It only lost a little bit off the end, so that probably won't materially affect the conclusions. Still, garbled assembly is the sort of thing that would cause a lot of your peers to dismiss the work out of hand as unreliable, so I would fix it.
Problem 2 with the GenBank file: It is horribly annotated. Use the HHpred server on at least the genes of the structural cassette and find out where the major structural proteins are encoded. You can do multiple databases at once at that site. I recommend pdb, cd, Pfam, PRK. For example, the gene said to encode capsid and scaffold protein, QYI86480, is really the tail fiber. This causes confusion in the paper. Firstly, no one tries to do classification with the most weirdly evolving protein in the phage. Secondly, the manuscript talks about calculating capsid structure and T numbers from capsid, but there is no gene associated with the word capsid throughout the manuscript except for the tail fiber. The major capsid protein is QYI86490. The GenBank file doesn't annotate it. It isn't till the supplement that we find out that someone on the project knew that QYI86490 was the major capsid protein. That then leaves confusion as to which of the genes was used in the network analysis, the tail fiber, or the actual scaffold and major capsid proteins. To compound the confusion, fig. 5 shows one gene marked as structure or morphogeneic in the vicinity of the MCP gene. I presume that was an attempt to mark MCP. But the adjacent scaffold protein wasn't marked, and the number of frames in that region on fig 5 doesn't match the GenBank file. That makes it hard to know which gene is which on the figure. You might consider marking MCP, portal, TMP on the upper track the way you marked terminase, transposase, and helicase on the lower track. The right way to label a gene in the GenBank file itself is to include the name of a family database accession. That can be put in a note or an evidence tag, or even as a live link. That allows the user to look up exactly what you matched, and even find a citation about it. If you just copy a phrase like "capsid and scaffold", it becomes impossible to track back how you, or whoever you copied it from, got that idea. For example, "capsid and scaffold protein" is an ontology term used to cover any gene vaguely in the structural gene module. So "capsid and scaffold" with an accession to an ontology group is vastly different than "capsid and scaffold" with the accession to the HK97 crystal structure HMM.
Problem 3 with the GenBank file: The phage taxonomy is listed as Caudoviricetes; Podoviridae
First, it's not a podovirus; it's a siphovirus. Your paper clearly shows that.
Secondly there is no Podoviridae or Siphoviridae in Caudoviricetes. Caudoviricetes was invented by ICTV to replace Caudovirales expressedly as part of their effort to erase Podoviridae and Siphoviridae from existence. I think that was a terrible idea, precisely because for a phage like yours, that removes all useful identifying information from their controlled vocabulary. However, NCBI is busily erasing Podoviridae and Siphoviridae from their database, bad idea or not.
The best workaround I can recommend is update the taxonomy string to:
Caudoviricetes; Unclassified, and add a note saying "Determined to be a siphovirus by electron microscopy". NCBI will eventually butcher your taxonomy string no matter what you put there, but presumably a note about experimental evidence will survive unscathed.
I strongly recommend that you submit an updated, repaired, GenBank entry. Otherwise, you're going to have to put a table in supplementary material giving enough of an annotation to make the paper readable and the calculations reproducible. It looks to me like your sequence will be the only annotated version of a very large family, so lots of people will be looking at this GenBank entry and comparing their phage sequence to it, and you don't want them seeing the quality that's there now.
In the paper itself:
This construct "consecutive series of nine BLAST searches" is confusing. As best I can tell it was blastp searches of tail fiber and large terminase, and blastn searches of whole genome, with matches compiled from several different databases. But I'm not completely sure. For each figure, say which proteins, or if it's whole nucleotide sequence; not how many blast searches you did.
line 52 reluctance to culturing techniques -> resistance to culturing techniques.
Line 116: this may be the first paper I've ever reviewed where the authors correctly identified this method as a phylogenomic analysis.
Line 149: The EM work is well done. There's no question about this being a siphovirus.
Line 247: "The ...terminase genes were not conserved". "Conserved" is the wrong word for what they are not. More correct would be to say that several different terminase families were represented in Vespunoviruses or that Vespunoviruses were not monophyletic for large terminase.
Fig 5. They did a good job with this figure by circularly permuting all the sequences the same way. This is the most useful figure in the paper. There is a small detail I notice. The gene downstream of portal is an SPP1 gp7 homolog, which is correctly annotated as a minor head protein in the GenBank file but not indicated as such on the figure. I notice it because I've recently had cause to become interested in this gene. It is not marked as conserved on fig. 5, but there doesn't seem to be the right number of frames in this interval. Just check what's going on in this region once more.
I presume that the authors understand that to get the name assigned, there is a form to get from the ICTV website and submit to them.
-------------------------------------------------------------------------------------------------
I think you should hold their feet to the fire to fix the GenBank submission. This is especially so since this is apparently going to be the only annotated sequence of a very large family. I've suggested some improvements to the manuscript, but I don't feel insistent about them.
Reviewer 2 Report
The manuscript by Pchelin et al. entitle “A genome of temperate Enterococcus bacteriophage placed in a space of pooled viral dark matter sequences” identified and characterized a potential new temperate phage VEsP-1. They showed the phenotype of the phage VEsP-1, and sequenced the genome of the phage, then they analyzed the genome and constructed its phylogenomic tree. The results are interesting to the readers of Viruses, but some other characteristics of the bacteriophage should be stressed.
1. The authors should start the results section with how did they get the new temperate phage, and why it is defined as a temperate phage?
2. Whether the phage can be integrated into the genome of the host after infection? If so, what conditions can induce its induction? Such as SOS response, UV.
3. Some basic information of the phage should be offered? Such as one step growth curve of the phage.
4. Some more stronger evidences should be offered to support the new genus “Vespunovirus”.